# Relevant Study: Patient and Clinician Perspectives on Clinically-Meaningful Outcomes in Advanced Pancreatic Cancer

**DOI:** 10.3390/cancers15030738

**Published:** 2023-01-25

**Authors:** Rille Pihlak, Melissa Frizziero, Soo Yit Gustin Mak, Christina Nuttall, Angela Lamarca, Richard A. Hubner, Juan W. Valle, Mairéad G. McNamara

**Affiliations:** 1Division of Cancer Sciences, University of Manchester, Manchester M20 4BX, UK; 2Department of Medical Oncology, The Christie NHS Foundation Trust, Manchester M20 4BX, UK

**Keywords:** pancreatic cancer, quality of life, patient views on treatment

## Abstract

**Simple Summary:**

Pancreatic cancer has a dismal prognosis and is one of the deadliest cancers globally. This prospective investigator-designed longitudinal questionnaire study aimed to evaluate expectations and priorities of patients with advanced pancreatic cancer and their clinicians. Results revealed that there is a mismatch between patient and physician views about the aims, priorities and expected benefit from the treatment for advanced pancreatic cancer. The main findings were that patients significantly overestimated the expected length of time extension that chemotherapy would offer, and when making decisions about treatment options: patients prioritised length of survival, while physicians thought that patients would prioritise the best balance between side-effects and survival. Overall, patients in this study had significantly higher hopes for treatment leading to life extension, compared to their physicians, and also had a lot of fear and worry about the future with poor symptom scores and quality of life.

**Abstract:**

Pancreatic ductal adenocarcinoma (PDAC) is an aggressive cancer with a poor prognosis and significant symptom burden. This prospective observational study aimed to evaluate expectations and priorities of patients with advanced PDAC and their clinicians through a study survey and two quality of life (QoL) questionnaires (QLQ-C30 and PAN26) at three time-points: baseline (T1), before (T2) and after (T3) their 1st on-treatment CT scan. Over a 1-year period, 106 patients were approached, 71 patients and 12 clinicians were recruited. Choosing between treatment options, patients prioritised: 54% overall survival (OS), 26% balance between side-effects and OS, 15% could not choose and 5% favoured symptom control. These were significantly different from the clinician’s answers (*p* < 0.001). Patients who prioritised OS had higher symptom burden (*p* = 0.03) and shorter OS compared to those who prioritised balance (*p* = 0.01). Most (86%) patients had personal goals they wanted to reach; clinicians knew of these in 12% of instances. Patient and clinicians’ views regarding survival improvement from chemotherapy were significantly different: 81% of clinicians and 12% of patients thought 1–2 or 3–6 months extension, 58% of patients and 0% physicians thought 1–5 or >5 years (*p* < 0.001). At T1, patients had low QoL and worst symptoms were: ‘Future worries’, ‘planning of activities’, fatigue and pain. Patients were willing to accept significantly higher amounts of side-effects as a trade-off for extra time, than clinicians thought (*p* < 0.001). Overall, there are significant discrepancies between patient and clinicians’ views about the aims, priorities and expected extension of life.

## 1. Introduction

Pancreatic ductal adenocarcinoma (PDAC) is one of the most fatal cancers [1]. Most patients have advanced disease (stage 3–4) at presentation [2], where average overall survival (OS) remains around 6 months [3], and recent clinical trials with either chemotherapy combinations or novel agents have proven limited survival benefit [4,5]. Therefore, PDAC is a major cause of cancer mortality, and it is estimated to become the 2nd leading cause of cancer-related death in the US by 2030 [6].

Additionally, health-related quality of life (HRQoL) has been found to be significantly worse than population norms amongst patients with PDAC based on the European Organisation for Research and Treatment of Cancer (EORTC) quality of life core questionnaire (QLQ-C30) [7]. Similar findings were reported in the QOLIXANE study that gathered real-life HRQoL and efficacy data of patients with metastatic PDAC starting first-line treatment [8], where patients were asked to complete a QLQ-C30 questionnaire at baseline and monthly during their treatment. This study showed worse scores for role functioning and fatigue and in a multivariable analysis, the physical functioning (PF) and nausea and vomiting (NV) scales had the most significant effect on OS [8]. Patients with PDAC have been found to have high prevalence of depression, 33–50% [9] and up to 78% in one study [10]. This can in turn lead to worse QoL, role-, emotional- and social functioning and worse fatigue, pain and appetite loss than patients without depression [10].

The short life expectancy and high symptomatic burden of patients with PDAC make QoL and symptom control a priority in their management. It also makes it critical to understand how patients perceive these outcomes; and what they and their clinicians consider to be a clinically meaningful benefit from treatment. Combination chemotherapy regimens used in PDAC can result in improved survival, but at the expense of significant side-effects [4,5], and little is known as to whether patients would have made a different choice after experiencing the side-effects of treatment.

In patients with various types of cancers, previous research has shown that patients self-reporting and dealing with their symptoms early, can lead to improvement in HRQoL [11] and even extend OS [12], and integration of early supportive care for patients with PDAC has demonstrated improvements in survival outcomes [13,14].

Compared to healthy volunteers, medical professionals or other groups, patients with cancer [15] have been shown to be more likely to choose chemotherapy, even with small survival benefits [16], more toxicities [17,18], irrespective of whether their QoL deteriorated or improved [19], and even if the clinicians presented the results pessimistically [20].

Whilst most patients with advanced cancer want to be given information about their prognosis and expected outcome of their treatment [21,22,23], various factors can influence patient choices. For example, QoL, anxiety and depression [10,24]; physician opinions not matching patients’ [25,26]; patients not knowing or understanding prognosis [27,28,29]; patients unrealistic views about treatment outcomes [30,31] and the impact it can have on their decision making [29,30]; and there have been mixed results reported in relation to choice between QoL and length of life [15,32,33,34,35].

No previous studies report on views of patients with advanced PDAC on their cancer diagnosis, treatment received and goals, and how these compare with those of their clinicians. Hence, the aim of the present study was to investigate this further and to determine what aspects were considered meaningful to patients.

## 2. Materials and Methods

This was a prospective investigator-designed longitudinal observational questionnaire study. The primary objective of the study was to evaluate patient and clinician views on pancreatic cancer diagnosis, treatments given, patient goals and meaningful outcomes. The secondary objective was to provide a descriptive analysis of the change in these views in relation to treatment response, side-effects and changes over time.

The study used a purposefully built questionnaire, together with the previously established QLQ-C30 [36] and its supplementary pancreatic cancer module PAN26 [37]. Key themes and topics from the published literature were reviewed to develop the questions in the survey, and QLQ-C30/PAN26 were used to assess HRQoL. The designed questionnaire was pretested before data collection to remove essential errors [38,39] by multiple expert reviews, field testing by patient representatives and ethics committee review. The study was approved by the Health Research Authority (HRA) and Health and Care Research Wales (HCRW) on the 16/05/2018 (reference 18/NW/0293).

Patients inclusion criteria was: (1) patients with newly-diagnosed advanced (unresectable) pancreatic adenocarcinoma who were seen at a single tertiary referral centre in the Hepatopancreaticobiliary (HPB) new patient clinic; (2) unresectable pancreatic cancer whose treatment aim was palliative; (3) satisfactory English language skills to fill in the study questionnaire by the participant themselves. Patients were excluded from the study if they were not fit enough for anticancer treatment or when surveillance was planned instead of anticancer treatment. Physicians or nurse clinicians who saw patients in the HPB clinic during the study period were eligible as clinician participants.

After informed consent, the following demographic information was collected from electronic patient records: date of birth, date of death (if applicable), age, gender, stage of disease, Eastern Cooperative Oncology Group (ECOG) performance status (PS) at baseline, chemotherapy received, outcome of the first treatment CT, documented discussion about prognosis (yes/no, limited to the first month after initial visit).

Patients were invited to complete the study survey and QLQ-C30/PAN26, at three time points (Figure 1):Time-point 1 (T1): Before starting palliative chemotherapy treatment.Time-point 2 (T2): After starting, and before first on-treatment CT scan (at least one dose of chemotherapy, ideally after 2–3 months).Time-point 3 (T3): After the first CT scan (after the patient has received the results of the CT scan).

A paired survey was completed by their corresponding clinician at each of these three time points. If the patients stopped treatment early and were not having a mid-treatment scan, they were also asked to complete these forms as T3.

All data from the surveys were entered into a purpose-built electronic database. All study documents were pseudoanonymised using a unique study ID.

### Statistical Methods

Summary statistics were provided for patient demographics. Changes in categorical variables with a binary scale and unordered categories were examined using the Fishers paired exact test. Changes in ordinal categorical variables and variables with a Likert scale were assessed using the Wilcoxon matched-pairs test. The QLQ scores were analysed using the paired t-test [40], as data were continuous and approximately normally distributed. The QLQ values were compared to the EORTC reference manual averaged scores for HPB cancers [41] and the EORTC Quality of Life Group thresholds for clinical importance (TCI) [42]. Previously, 10-point changes in the QLQ-C30 have been found to be clinically meaningful and correspond to supportive care needs [43], thus, this cut-off was used.

The sample size of this study was not based on statistical power as it was not aimed at analysing survival differences. All sequential eligible patients were approached to avoid bias.

Statistically significant results were defined as having a *p*-value < 0.05. The IBM SPSS Statistics software package (version 23) was used for data analysis.

## 3. Results

### 3.1. Study Accrual

The study was opened in May 2018, closed in May 2019, and follow up was completed in August 2019. During the one-year recruitment period, 106 patients were approached, 35 declined study entry or decided not to start chemotherapy. Full accrual and each patient’s length on study are illustrated in Figure A1 and in the CONSORT diagram in Figure 2. Twelve clinician participants were eligible and all consented.

There was an average of 71 days (2.3 months) between T1 and T2, 112 days (3.7 months) between T1 and T3 and 40 days (1.3 months) between T2 and T3. To assess changes between T1–T2 and T1–T3, only the 39 patients who filled in both surveys for T2 and the 36 patients for T3 were included in the analysis.

### 3.2. Patient Characteristics

Patient baseline characteristics are further described in Table 1. Around half (51%) of all patients had a prognosis discussion documented at their first visit, or during the first month of treatment.

### 3.3. Clinician Characteristics

Of the 12 clinicians that were consented for the study, four were medical oncology senior faculty, six junior doctors, one nurse clinician and one General Practitioner (GP).

### 3.4. Survey Findings

#### 3.4.1. Involvement in Decision Making

As illustrated in Figure A2, when asked about the extent to which patients wish to be involved in decisions regarding their treatment, most patients answered at T1 that they preferred sharing responsibility with their doctor and there was no significant change between time points.

#### 3.4.2. Impact of Treatment on Patients’ Everyday Lives

For 68% of patients, it took up to 1 hour to come to the hospital for treatment (one way), for 26% it took 1–2 h. The majority (79%) had to come 2–3 times a month and most (95%) drove themselves or had others drive them. Patients were asked how the treatment affected them financially (Figure 3) and 29% of patients were a little or a lot out of pocket at T1; this increased to 41% at T2 (*p* = 0.036) and 42% at T3 (*p* = 0.034).

#### 3.4.3. Expectations for Treatment

When asked about the aim of treatment (Figure 4A), the majority of patients thought it was to keep the cancer under control and manage symptoms or end-of-life treatment, with a few patients still thinking that the aim was cure, 13 (15%) thought that it was to shrink the cancer to make it surgically resectable (including nine patients with metastatic and four with locally advanced PDAC).

Compared to clinicians, there was a statistically significant difference (*p* < 0.001) in the answers to this question, where all (100%) of clinicians responded that the goal of treatment was to control the cancer and manage symptoms. There was no significant change between the time points in patient or clinician views.

Patients were then asked about the percentage of patients they think will have serious (life-threatening or requiring hospitalisation) side-effects with the same type of chemotherapy that they are starting, and 32.3% expected it to be 6–10%, 21.5% expected 1–5%, 20% expected 11–20% and 18.5% expected >20%. Similarly, when asked about the likelihood of chemotherapy reducing their current cancer symptoms (Figure 4B), 40% answered that this was very or completely likely, and there was no significant change between the time points in this question even when patients had started chemotherapy (T2 and T3).

Patients and clinicians were also asked about the likelihood of chemotherapy curing the cancer (Figure 4C) and the likelihood of cancer to respond to chemotherapy (Figure 4D).

Compared to clinicians, patients felt that there was a higher likelihood of chemotherapy curing the cancer, when 94% of clinicians reported that it was not at all likely, compared to 58% of patients (*p* = 0.02). Additionally, patients were more optimistic about cancer responding to chemotherapy (39% compared to 6% completely or very likely, *p* = 0.001) and prolonging their life (45% compared to 9% very or completely likely, *p* < 0.001). There was no significant change in these views between the time points. The majority (86%) of patients answered that they had personal or family goals that they would like to reach with the help of treatment. Spending time with family, self-care for as long as possible and being able to socialise were ranked as the top three priorities for patients, respectively (Figure 4E). The only statistically significant change between T1 and T3 was that spending time with family became more important to patients (*p* = 0.02).

Compared to clinicians, there was a significant difference in patients’ personal goals; 86% of patients indicated that they had goals they wanted to reach with the chemotherapy, whilst only 12% of clinicians were aware of these (*p* < 0.001). The importance of some goals varied between the two groups where being able to travel (mean 3.7 compared to 5.1, for patients and clinicians, respectively, *p* < 0.001), spending time with family (mean 1.4 compared to 2.4, *p* < 0.001) and special events (mean 3.2 compared to 4.7, *p* < 0.001) were all rated as significantly more important by patients than clinicians (lower numbers indicating more importance).

#### 3.4.4. Priorities When Choosing between Treatment Options

Most (54%) patients ranked longest survival as their main priority when making decisions about different treatment options. There were statistically significant differences between patient and clinician answers, where over half (59%) of clinicians indicated that the balance between side-effects and survival would be the main priority for the patient when choosing between treatment options, whilst longest survival was the most common response from patients (54%, *p* < 0.001, Figure 5). For both patients and clinicians, there were no statistically significant changes between time points.

On review of responses from patients who prioritised survival over balance, they had higher symptomatic burden (*p* = 0.03), more issues with constipation (*p* = 0.03), more appetite loss (*p* = 0.01) and borderline significantly worse role functioning (*p* = 0.058) at baseline. Comparing patient groups, based on their top choice in this question at T1, there was also a significant difference in OS (*p* = 0.01), where patients who prioritised symptom control, lived an average of 2.8 months from T1, patients who prioritised survival: 6.4 months, could not choose: 8.7 months, and who prioritised balance, lived 9.2 months (Figure A4).

#### 3.4.5. Survival Expectations

Patients and clinicians were also asked about their expectations of treatment for extending life (Figure 6A). Most patients (58%) expected the chemotherapy to extend their life by 1–5 or more than 5 years (46% and 12%, respectively). Similarly, when asked what would be the minimal extra time that would be important to them, 43% answered 1–5 or >5 years (32% and 11%, respectively) (Figure 6B). Comparing T2 answers to T1, there was an increase in patients (37% to 54%) reporting 1–5 or >5 years as the minimally important time acceptable, and also a slight increase in the time patients expected the chemotherapy to extend their life by (63% compared to 74% of patients expecting 1–5 or >5 years). However, both these results were not statistically significant.

There were significant differences between patient and clinician responses regarding the length of time chemotherapy was expected to extend patients’ lives (*p* < 0.001, Figure 6A) and the minimal extra time patients would consider to be important (*p* < 0.001, Figure 6B).

There were 12 (18%) separate patients who at any time point expected chemotherapy to extend their lives by >5 years, or who answered that >5 years would be the minimally important time for them.

Compared to other patients, these 12 had less problems with NV on the QLQ-C30 scale (*p* = 0.006) and borderline significantly (*p* = 0.05) higher summary score indicating less symptomatic burden. They were also more likely to choose survival length over balance between side-effects and survival as their main priority when choosing between treatment options, but this was borderline significant (*p* = 0.054). There was no difference in willingness to accept large amounts of side-effects between these patient groups and no significant difference in overall survival.

#### 3.4.6. Side-Effects Trade off

Patients were then asked how many side-effects they were willing to endure for the minimally important time they answered in the previous question, and almost all of them were willing to accept small or medium amounts of side-effects (defined as taking oral medication for side-effects at home), whilst 57% were not willing to tolerate large amounts of side-effects (eight did not answer) (Figure A3). If the minimally important time was doubled, more patients were willing to bear large amounts of side-effects (47% increased to 60%). Two-thirds (67%) of patients were willing to accept chemotherapy if it controlled symptoms of their cancer but did not extend survival.

Between the time points, the only statistically significant change was a decrease from 70% to 43% (*p* = 0.04) of patients who would be willing to accept chemotherapy with a large amount of side-effects, if the minimally significant survival time was doubled (between T1 and T2).

Comparing patient and clinician answers (Figure A3), there was a statistically significant difference in patient willingness to accept chemotherapy with medium and large amounts of side-effects as a trade-off for minimal important time gain (100% compared to 88% for medium (*p* = 0.02) and 47% compared to 9% for large amount (*p* < 0.001), in patients and clinicians, respectively).

Clinicians underestimated patient willingness to accept chemotherapy if it controlled symptoms of cancer but did not extend survival (67% patients compared to 46% clinicians, *p* = 0.05).

#### 3.4.7. EORTC QLQ-C30 and PAN26

Of a possible 100 points, Table 2 details all the mean results from the QLQ-C30 and PAN26 scales at baseline and changes between the time points; these are further illustrated in Figure 7. At least 10-point (*p*) changes between T1, T2 and T3 are highlighted in Table 2.

Compared to the average scores for HPB cancers in the EORTC reference manual [41], scale scores with at least 10p difference were worse: pain (at T1 44p vs 29.6p), appetite loss (at T1/T2 47p vs 32.3p), constipation (at T1 30p vs 20p) role functioning (at T2 50 vs 65.2), social functioning (at T2 55 vs 69), fatigue (at T2 52 vs 41.2), NV at T2 30 vs 12.4, pain (at T2 39 vs 29), insomnia (at T2 43 vs 32), diarrhoea (at T2/T3 30 vs 11.1) in all current study patients versus EORTC HPB standard, respectively.

Compared to EORTC TCI [42], patients in the current study had worse PF (72.7, 65.5, 70.5 compared to 83), fatigue (45.8, 51.9, 44.6 compared to 39), pain (44.4, 38.9, 26.3 compared to 25), NV (18.6, 29.8, 15.6 compared to 8) and diarrhoea (23.2, 30.2, 21.4 compared to 17) in all three time points (T1, T2, T3, respectively). At T2, they also had worse role functioning (50.0 compared to 58), social functioning (54 compared to 58), emotional functioning (67 compared to 71), appetite loss (55.6 compared to 50), dyspnoea (20.2 at T2 and 18.3 at T3 compared to 17) and financial difficulties (18.1 at T1 and 18.2 at T2 compared to 17).

Sixty one percent of patients died during the study timeframe, and for those patients, the median OS from T1 was 3 months (range 0-13 months). At the time of final analysis, 93% of patients had died; the median OS for all patients was 7.39 months (95% CI 5.3–9.5 months). Overall survival based on first line chemotherapy is illustrated in Figure A5.

## 4. Discussion

Similar to previous research [44,45,46], patients in the current study wanted to be involved in decision making about their treatment, and the diagnosis of cancer and treatment had a significant impact on patient’s everyday lives, especially the burden of financial toxicities, which significantly worsened over time. The patients’ most important goals were spending time with family and self-care for as long as possible, but a significant decline in role functioning became evident at later time points, highlighting that patients might not be able to do things that are most important to them, due to symptoms and cancer treatment.

In the current study, 58% of patients were aware that the treatment was unlikely to cure their cancer, whilst 17% thought it would be slightly likely and 8% moderately likely, which is more than the previous study by Weeks et al. [30] of patients with metastatic lung or colorectal cancers. There were still 2% of patients who thought that the aim of treatment would be to cure the cancer, which is similar to 5% reported in another study by Loh et al. [31].

Most patients also expected <10% would experience serious side-effects from their treatment, whilst all treatment regimens that these patients received produced >10% Grade 3–4 side-effects in previous clinical trials [4,5]. Compared to clinicians, patients were also significantly more optimistic about the effectiveness of treatment.

Most patients had personal or family goals that they wanted to reach with the help of treatment, and only a small number of clinicians were aware of these. This difference could be due to patients not always communicating this to their clinician, or this could also be a result of a lack of communication and the time pressures associated with busy clinics. Discussion regarding patient goals is important, as previous research has shown that meaningful life events and relationships are sources of hope for patients [21]. Patients who reach their goals have less anxiety [24], especially as spending time with family became even more important to patients over time in the current study.

One of the main differences between clinician and patient expectations in this study was that whilst the majority of patients indicated OS as their main priority when choosing between chemotherapies, clinicians thought that the patients would prioritise the balance between OS and side-effects. This discrepancy is especially interesting as most patients still answered in a later question that they were willing to undergo treatment if it controlled symptoms of cancer but did not lead to longer survival. The patients who prioritised survival seemed to be the ones that were already struggling with worse symptoms and had a worse OS compared to patients who prioritised the balance between side-effects and survival. This could be due to patients already feeling worse at baseline and desperately wanting more time.

As previous research into patient choice between length of life and QoL revealed mixed results [32], it is particularly interesting that in this study, the two other answer options: *least amount of side-effects* and *controls the symptoms of my cancer*, were all together chosen only by 5% of patients. Whilst previous research [16,32,35,47] has shown that most patients thought both survival and QoL as important, it was expected that in the current study, the option *best balance between side-effects and survival* would be chosen as the main priority. However, even given the option of choosing both survival and QoL, patients still prioritised length of life. This could be due to the fact that for these patients, the expected average prognosis would be measured in months, and highlights the fear that patients have of running out of time and therefore they focus on OS. As patients wanted to be an equal partner in decision making, the results from this question indicate that clinicians and patients have different priorities in mind for patients.

The second main finding of this study is the differences between patient and clinician views in relation to prognosis and minimal important survival time gain. Most patients expected life extension to be 1–5 or >5 years, and whilst half of the patients had documented discussions about prognosis, this did not impact their expectations. Clinicians who are more aware of the expected prognosis averages from previous clinical trials all expected the extension of life to be less than 1 year (Figure 6). This discrepancy between patient and clinician expectations could have implications for patient decision making. In the study by Weeks et al. [29], patients who expected their survival to be more than 6 months favoured life-extending therapy over best supportive care and were more likely to undergo aggressive treatment, but their 6-month survival was not actually longer. The patients most likely to choose life-extending therapy were the ones where the mismatch between clinician and patient expectations of length of survival was the widest [29]. This was similar to the current study, where most clinician answers expected the life extension from chemotherapy to be up to 6 months, and only small percentages of patients responded similarly. The patients who expected treatment to prolong their survival by >5 years or that the minimally important time for them would be >5 years, were more likely to choose survival as their primary aim, but did not have longer OS. Thus, similarly to the Weeks at al. [29] study, patients with the widest mismatch in expected length of life did not live longer, but were more likely to choose survival as their treatment priority.

Large discrepancies between expectations of benefit and experienced benefit have been previously linked to feeling more anxiety and depression in patients with ovarian cancer in a study by Sjoquist et al. [24]. In the current study, the QLQ-C30 *emotional function* scale and PAN26 *future worries* and *future planning* scales showed that on average, most patients in the current study had issues with emotions and worry. At the same time, most patients died during the study despite expecting their survival to be 1–5 or >5 years. Therefore, the link between large discrepancy in expectations versus reality and fear and worry can also be seen in almost all patients in the current study.

In the previously mentioned study by Sjoquist et al., they proposed that clinicians should encourage realistic hope, targeted toward achievable goals [24]. As depression has been shown to be a major problem in patients with PDAC [9], it might be important to have candid conversations with patients rather than give false hope that could in turn lead to further psychological harm. Giving prognostic information in a realistic and open way, tailored to the individual, has been shown to be viewed as hope-giving by patients [48], and that hope is maintained even when the news is bad [49]. In the current study, it was found that around half of patients had documented prognosis discussions within the first month of their palliative treatment. Whilst we do not know if this was done but just not documented, it does highlight the need to remind clinicians of the importance of prognostic discussions, especially as patients who become aware of their terminal status by worsening condition, or by chance, have been previously found to have worse QoL [50].

In keeping with previous studies [16], in the current study, most patients were willing to accept greater amounts of side-effects, especially when the hypothetical time was doubled. They were also willing to accept treatment when it would control their symptoms but would not extend their survival. Potentially due to being more aware of the risks and mortality associated with treatments, clinicians underestimated the number of patients who would be willing to accept side-effects or treatment if it did not extend survival.

At all time points, patients had significant issues with symptoms, functions and quality of life, and there were also significant changes over time, especially at T2, where many of these worsened. These results indicate the high symptomatic burden of patients recently diagnosed with advanced PDAC and how these worsen during first line treatment. As >10 p changes in any scales have previously been associated with increased supportive care needs [43], the study also shows that most patients would be likely to require involvement of specialist supportive care services. Early and systematic supportive care consultations have demonstrated improvements in survival and QoL outcomes in patients with PDAC [13,14], therefore highlighting the need for monitoring patients’ QoL scales continuously, which may enable detection of more subtle, but significant, changes over time in symptoms [8,51].

To the best of the author’s knowledge, this is the first study to link patients QoL scores with their views about treatment. The patients who prioritised OS (over balance between QoL and OS), had worse symptomatic burden, but ended up having, on average, a shorter OS. This could also be an indication that patients who are symptomatically worse at baseline, have a fear of running out of time and would be willing to take any treatment that could prolong their life. Taken together with the previously reported prognostic value of baseline QoL scales on OS of patients with PDAC [8,51], this highlights that baseline QoL questionnaires could be used to help decision making, guiding discussions with patients about treatment aims and highlighting the importance of symptom control.

Previous research has shown that patients might be able to cope with realism more over time [48], whilst in the current study, no significant reductions were noted in patient views about their prognosis. On the contrary, there was a trend towards patients expecting the chemotherapy to extend their life longer at T3 than in the T1. This could reflect the positive bias in the current results, as only 51% of patients reached the third time point, and all of those had stability or response on CT imaging. At the same time, the OS in the previous trials [51] was still much longer (19–30 months) than in the current one, and it is not known if “urgency” plays an additional role in patient decision making.

## 5. Study Limitations

A large number of patients deteriorated or died, mainly between T1 and T2, and this drop-out rate could be causing an inherent positive survivor bias in the study results. Unfortunately, rapid deterioration and drop-out is a common problem in the setting of advanced pancreatic cancer, as also seen in the QOLIXANE study [8]. In that study, only around a half of the patients filled in the month 3 QLQ-C30 questionnaire [8]. In the current study, the patients who did not reach another time point had worse PF, less aggressive treatment at baseline and died prior to the average time other patients filled in the T2 questionnaire (2.17 compared to 2.3 months). Future studies could include shorter intervals between surveys and potentially introduce a caregiver view from the beginning. Another potential limitation of this study was that it included 83 questions (27 from the developed survey and 56 from QLQ-C30/PAN26) and this might lead to questionnaire fatigue [38,39]. In future studies, the questions should be limited to include relevant areas from this survey.

As the 12 clinicians recruited to the study had various levels of experience with managing patients with pancreatic cancer, there could be differences in their responses. However, whilst detailed reliable statistical analysis of clinician characteristics is not possible due to the small numbers, no clear descriptive differences between clinician answers were seen.

## 6. Conclusions

The results of this study demonstrate that there are important discrepancies between patient and clinician views. Patients have personal goals that they want to reach, they are more focused on length of survival and are willing to accept more treatment related toxicities to achieve that. At the same time, OS in patients with advanced PDAC continues to be short, symptomatic burden is high, including significant worries and anxiety, and treatment expectations do not often match their clinicians. These differences between patient and clinician views have clinical implications, therefore highlighting these and dealing with them at an early stage, could lead to improved patient outcomes. The findings of this study will educate the treating clinical community as to the importance of establishing patient goals of care and priorities at the beginning and during their treatment, and also highlights the need to offer psychosocial support, and have candid conversations, to ease the emotional impact of cancer and its treatment.

## Figures and Tables

**Figure 1 cancers-15-00738-f001:**
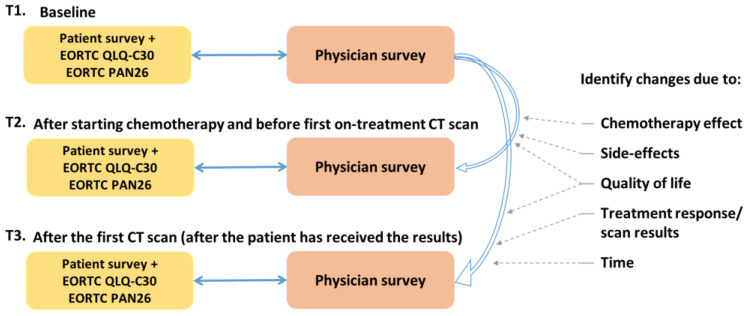
RELEVANT study outline.

**Figure 2 cancers-15-00738-f002:**
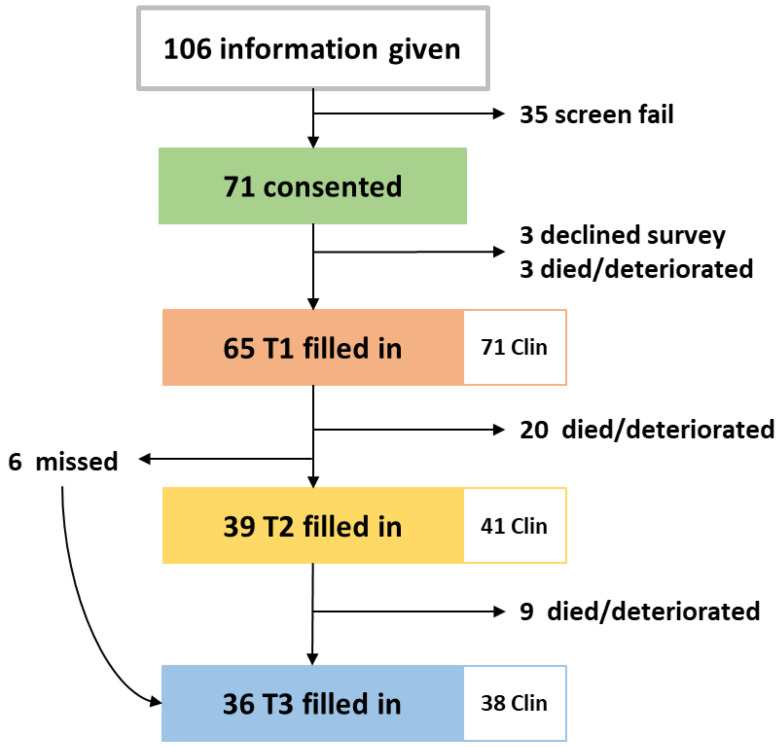
CONSORT diagram of patients included in the RELEVANT study and the paired clinician surveys filled in. Clin: clinician surveys.

**Figure 3 cancers-15-00738-f003:**
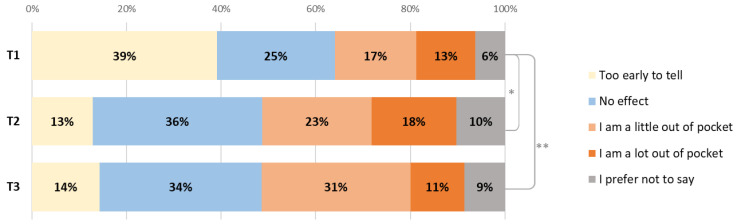
Changes in the financial impact of diagnosis or treatment on patients between T1, T2 (* *p* = 0.036) and T3 (** *p* = 0.034).

**Figure 4 cancers-15-00738-f004:**
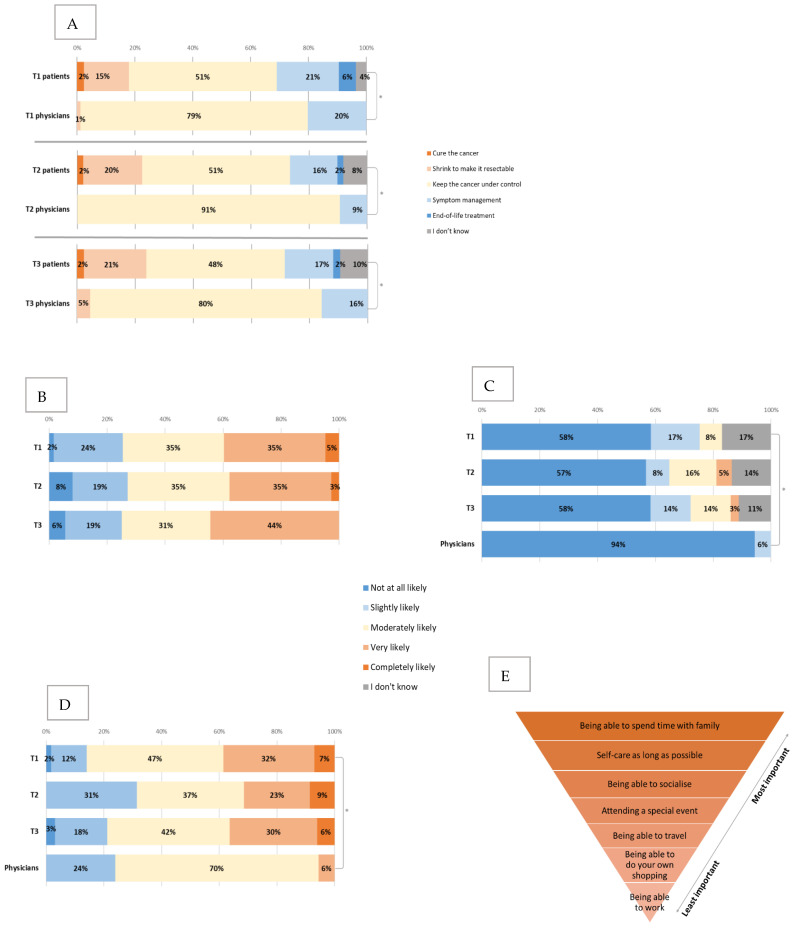
Expectations and priorities for treatment. (**A**) Changes in the patient and physician views about aims of treatment between T1, T2 and T3. * all *p* < 0.001 (**B**) Changes in the patients’ views about the likelihood of chemotherapy reducing their current symptoms between T1, T2 and T3. (**C**) Changes in the patients’ views about the likelihood of the treatment to cure the cancer between T1, T2 and T3 (compared to physician baseline views). * *p* = 0.02. (**D**) Changes in the patients’ views about the likelihood of the cancer to respond to chemotherapy between T1, T2 and T3 (compared to physician baseline views). * *p*=0.001. (**E**) Patients priorities ranked from most important (nr 1) to least (nr 7) important (baseline).

**Figure 5 cancers-15-00738-f005:**
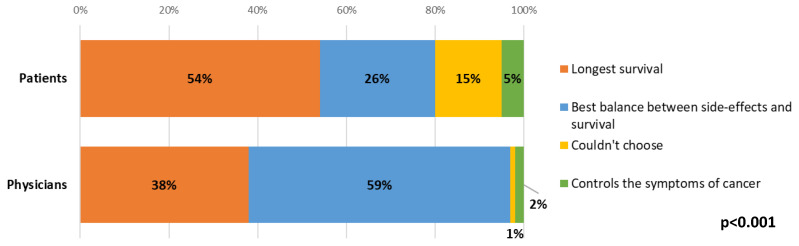
Differences between patient and clinician views about patient priorities when choosing between treatment options.

**Figure 6 cancers-15-00738-f006:**
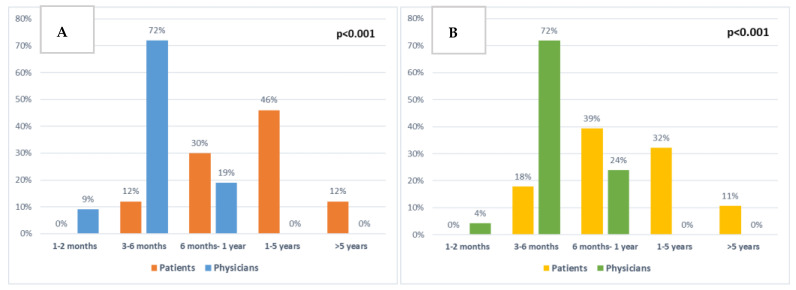
Comparison between patient and clinician expectations: (**A**) About chemotherapy extending patient survival at baseline (*p* < 0.001). (**B**) About minimal extra survival time that would be important to patients at baseline (*p* < 0.001).

**Figure 7 cancers-15-00738-f007:**
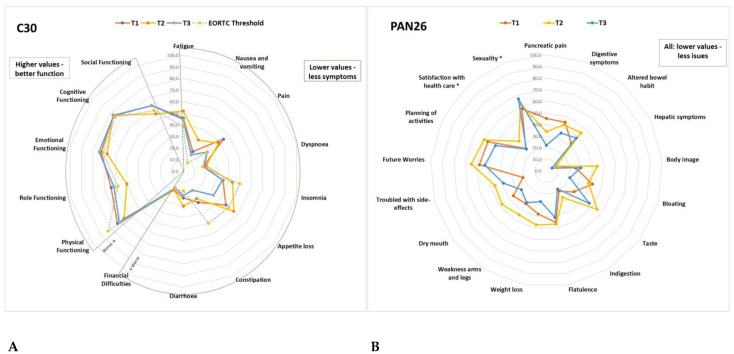
(**A**) Average QLQ-C30 scale scores at T1, T2, T3 compared to the EORTC Threshold for clinical importance. (**B**) Average PAN26 scale scores at T1, T2, T3. Details of the values and changes between time points are described in Table 2. *: function scales were switched for all scales to have the same direction.

**Table 1 cancers-15-00738-t001:** Patient participant baseline characteristics.

Baseline Characteristics	Number	Percentage
**Gender**		
Male	37	52%
Female	34	48%
**Age**		
(median, range)	65	(43–83)
**Stage**		
I-II unresectable	5	7%
III unresectable	10	14%
IV	56	79%
**ECOG PS**		
0–1	54	76%
2	17	24%
**Treatment started**		
Monotherapy (gemcitabine)	16	23%
Doublet chemotherapy (gemcitabine + nab-paclitaxel or capecitabine)	27	38%
Triple chemotherapy (FOLFIRINOX)	23	32%
No treatment due to rapid deterioration	5	7%

ECOG PS: Eastern Cooperative Oncology Group performance status; FOLFIRINOX- 5-Fluorouracil, Leucovorin, Irinotecan and Oxaliplatin combination chemotherapy.

**Table 2 cancers-15-00738-t002:** Average scores (out of 100) and changes between time points of QLQ-C30 and PAN26 scales of all patients.

		T1 Mean, 95% CI	T2 Mean, 95% CI	Change T1–T2 *	*p*-Value *	T3 Mean, 95% CI	Change T1–T3 *	*p*-Value *
**Global health status/QL (higher better)**								
**C30**	Global health status/QL scale	57.3 (52.0–62.7)	48.7 (40.2–57.2)	−9	0.11	63.6 (55.5–71.8)	2	0.71
**Functional Scales (higher better)**								
**C30**	Physical functioning	72.7 (67.0–78.3)	65.5 (56.1–75.8)	**−10 ↓**	0.11	70.5 (62.6–78.4)	−9	0.09
	Role functioning scale	63.8 (55.6–72.1)	50.0 (38.5–61.5)	**−17 ↓**	**0.039**	61.3 (49.5–73.0)	−6	0.47
	Emotional functioning scale	72.7 (66.8–78.7)	67.4 (56.9–78.0)	−2	0.76	74.4 (65.6–83.3)	4	0.45
	Cognitive functioning scale	77.1 (71.8–82.4)	77.6 (66.6–88.6)	−2	0.796	76.7 (66.9–86.4)	−2	0.76
	Social functioning scale	62.7 (55.3–70.1)	54.7 (43.6–65.8)	−7	0.316	62.9 (53.2–72.6)	−3	0.7
**PAN26**	Satisfaction with health care	73.9 (66.2–81.5)	65.3 (54.5–76.0)	−9	0.192	74.5 (63.9–85.1)	1	0.9
	Sexuality	42.4 (30.3–54.4)	38.2 (25.0–51.3)	−4	0.635	33.3 (19.5–47.2)	−9	0.32
**Symptom scales/items (lower better)**								
**C30**	Fatigue	45.8 (38.9–52.6)	51.9 (42.0–61.7)	9	0.197	44.6 (35.7–53.6)	4	0.55
	Nausea and vomiting	18.6 (12.2–25.1)	29.8 (18.0–41.6)	**12 ↓**	0.11	15.6 (7.6–23.6)	1	0.93
	Pain	44.4 (35.9–52.8)	38.9 (25.9–51.9)	-6	0.51	26.3 (15.9–36.8)	**−16 ↑**	0.039
	Dyspnoea	17.5 (11.6–23.4)	20.2 (8.8–31.6)	3	0.67	18.3 (10.0–26.5)	2	0.72
	Insomnia	35.0 (27.1–43.0)	43.4 (30.7–56.1)	9	0.26	34.4 (20.8–48.0)	−3	0.69
	Appetite loss	46.9 (37.8–56.0)	55.6 (41.5–69.6)	**10 ↓**	0.29	33.3 (21.5–45.1)	−5	0.52
	Constipation	29.9 (21.8–38.1)	26.3 (14.9–37.6)	0	1	18.3 (8.4–28.2)	**−14 ↑**	0.06
	Diarrhoea	23.2 (15.4–30.9)	30.2 (19.5–40.9)	4	0.61	21.1 (10.5–31.7)	−3	0.73
	Financial difficulties	18.1 (11.4–24.8)	18.2 (8.3–28.0)	3	0.63	16.1 (7.3–25.0)	0	1
**PAN 26**	Pancreatic pain	45.2 (38.5–52.0)	33.9 (24.3–43.4)	**−11 ↑**	0.054	22.0 (14.7–29.3)	**−23 ↑**	**<0.001**
	Bloating	41.2 (33.0–49.5)	37.8 (26.4–49.3)	−3	0.628	21.1 (11.6–30.6)	**−20 ↑**	**0.002**
	Digestive symptoms	44.9 (36.5–53.3)	42.6 (32.5–52.7)	−2	0.723	34.9 (22.7–47.2)	**−10 ↑**	0.179
	Taste	29.9 (21.3–38.3)	55.0 (42.1–67.8)	**25 ↓**	**0.002**	46.2 (35.0–57.5)	**16 ↓**	**0.023**
	Indigestion	20.7 (14.2–27.2)	26.9 (15.5–38.2)	6	0.346	18.3 (9.4–27.1)	−2	0.658
	Flatulence	45.2 (36.9–53.5)	46.7 (33.0–60.3)	1	0.853	40.9 (28.3–53.4)	−4	0.561
	Weight loss	37.9 (28.7–47.0)	47.2 (36.0–58.5)	9	0.197	26.9 (14.5–39.3)	**−11 ↑**	0.154
	Weakness arms and legs	33.3 (26.3–40.4)	44.4 (33.0–55.9)	**11 ↓**	0.099	32.3 (23.6–40.9)	−1	0.846
	Dry mouth	35.6 (27.9–43.3)	47.7 (35.0–60.5)	**12 ↓**	0.103	26.9 (16.7–37.1)	−9	0.172
	Hepatic symptoms	9.6 (6.2–13.0)	9.3 (4.3–14.2)	0	0.908	5.4 (1.7–9.0)	−4	0.093
	Altered bowel habit	31.9 (24.8–39.1)	44.4 (33.6–55.3)	**13 ↓**	0.056	38.2 (28.3–48.1)	6	0.302
	Body image	25.7 (18.4–33.0)	44.0 (32.1–55.9)	**18 ↓**	**0.010**	29.8 (19.0–40.6)	4	0.528
	Troubled with side-effects	20.9 (13.0–28.9)	46.1 (35.7–56.4)	**25 ↓**	**<0.001**	38.4 (28.1–48.7)	**17 ↓**	**0.008**
	Future Worries	57.9 (48.8–67.0)	64.8 (53.4–76.2)	7	0.341	53.1 (41.0–65.3)	−5	0.526
	Planning of activities	56.3 (46.7–65.9)	60.0 (48.6–71.4)	4	0.619	49.0 (36.0–61.9)	−7	0.358

* Changes calculated only between patients who filled in both questionnaires (39 patients T1-T2, 36 patients T1-T3). Time point means include all patients who filled in the questionnaire at that time point (65 patients at T1, 39 patients at T2, 36 patients at T3). CI: confidence interval ** ↑**: clinically significant improvement by at least 10 points. ** ↓**: clinically significant worsening by at least 10 points. Numbers in bold indicate either statistical or clinical significance.

## Data Availability

The data presented in this study are available in this article.

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
