# Peer review of "Relevant Study: Patient and Clinician Perspectives on Clinically-Meaningful Outcomes in Advanced Pancreatic Cancer"

_cancers, 2023, doi:10.3390/cancers15030738_

Round 1
Reviewer 1 Report
This study aimed to evaluate expectations and priorities of patients with advanced PDAC and their clinicians through a study survey and two quality of life (QoL) questionnaires (QLQ-C30 and PAN26) at 3 time-points. They found that patients in this study had significantly higher hopes for treatment leading to life extension, compared to their physicians, and also had a lot of fear and worry about the future with poor symptom scores and quality of life.
Specific suggestions:
1. It is suggested to display the basic information and clinical pathological information of the included patients in chart form.
2. It is suggested that the author should provide the pathological diagnosis details of included patients; provide the specific number of cases in each TNM stage; and provide typical CT images of patients in different stages at T1, T2 and T3 time-point.
3. It is suggested to provide the chemotherapy protocols received by included patients. Different chemotherapy protocols might have different clinical effects and side effects on patients.
4. It is recommended to adjust the sequence of Figure 4 according to the sequence described in the results section.
5. For the treatment options, was there any difference in the choice of T1, T2 and T3 time-point between patients and clinicians? In addition, it is suggested to show the effect of treatment options on survival in the form of survival curve.
6. Figure Appendix 3 does not intuitively display the related results of Side-effects trade off.
7. Some results were simply described in the results section, not shown in the form of figures or tables, such as page 6, line 225-231; page 8, line 269-277; page 9, line 283-292。
8. In this study, a large number of patients deteriorated or died during the study, and similar phenomena might cause significant deviation to the results. Was this phenomenon related to the lack of strict inclusion and exclusion criteria? In addition, it is suggested to provide more detailed inclusion and exclusion criteria in the method section.
9. What is the clinical value of the these results for the clinical diagnosis and treatment of patients with advanced pancreatic cancer?
Author Response
1-3: A table summarising patient baseline characteristics, including TNM stage and choice of chemotherapy regimen, was added. Histopathology was not included in this table as this was adenocarcinoma for all patients, which is included in the study inclusion criteria within the methods section. Typical CT images of patients were not added as these were considered outside the scope of this paper.
Sequence in Figure 4 was adjusted according to the results section.
- A clarification was added to section 3.4.4 that there was no difference between time-points for either patients or clinicians. Survival curve according to the chemotherapy strategy received by patients was added (Figure Appendix 5).
- Figure Appendix 3 was amended to be clearer and easier to understand the differences between patients and physicians.
- Inclusion of all data in both text and figure format was avoided to prevent duplication and confusion, and additional figures would not add to the messages covered.
- As discussed under study limitations, unfortunately as these are patients receiving 1st line treatment for advanced pancreatic cancer, their deterioration or death during study is relatively typical for the clinical course of this aggressive cancer and similar drop-out rates are seen in other observational studies with patients with advanced pancreatic cancer (for example the referenced QOLIXANE study). Further details about the inclusion and exclusion criteria was added to the methods section.
- We believe this study provides valuable insight into what is important to patients with advanced pancreatic cancer and emphasises the poor QoL and high symptom burden these patients have. Both of which can be used to guide candid discussions with these patients about the goals, aims and expected outcomes of treatment. As these differences between patient and clinician views have clinical implications, highlighting these and dealing with them at an early timepoint, could lead to improved patient outcomes.
Reviewer 2 Report
This manuscript aims to address the importance of baseline Quality of life questionnaires and how it can play a role towards decision making, guiding discussions with patients about treatment aims and highlighting the importance of symptom control.
Major Comments:
- A table with the patient characteristics (age, sex, health background information, etc.) must be included in the manuscript.
- On line 259, the authors mentioned that – “Comparing patient groups, based on their top choice in this question at T1, there was also a significant difference in OS (p=0.01), where patients who prioritised symptom control, lived an average of 2.8 months from T1, patients who prioritised survival: 6.4 months, couldn’t choose: 8.7 months, and who prioritised balance, lived 9.2 months.” Where is the data for actual survival in the manuscript? Instead of just mentioning herein, it would be nice to see a figure with this information. Also, it would be interesting to see how such patients whose priority, for example was survival, correlated to both treatment response to chemotherapy, etc.
- On line 173, the authors state that – “12 clinicians that were consented for the study, 4 were medical oncology senior faculty, 6 junior doctors, 1 nurse clinician and 1 General Practitioner (GP).” The authors do not address that the results of the questionnaires could be influenced by the variability in “Clinician characteristics”.
Minor Comments:
- Correct grammar and spelling throughout. For example, correct the spelling – prioritized.
- The authors need to correct the writing in Figure 6 legends. For example, they state that – “Figure 6. Comparison between patient and clinician expectations: A. about chemotherapy 279 extending patient survival at baseline (p<0.001). B. about minimal extra survival time that would be 280 important to patients at baseline (p<0.001).” After “A.”, “about” should be written as About and so on…
Author Response
- A baseline characteristics table has been added.
- Kaplan-Meier curves for OS according to patient choice at T1 were added as Figure Appendix 4. In relation to the association with response to treatment, unfortunately only 54% of patients made it to T3 where the first CT scan was done to assess treatment response (29 patients had stable disease and 6 patients had partial response on the scan). Therefore, the small number of patients does not allow for a reliable statistical analysis of the correlation between response and patient choice in this question.
- The small number of clinicians does not allow reliable statistical analysis of clinician characteristics, but no clear differences were seen between clinician responses. Details about this was added to the study limitations section.
- The manuscript was written in UK English, if this needs to be changed to US spelling, I am happy to do so. Manuscript has been re-reviewed by co-authors who are native English speakers.
- Figure 6 legend was amended as suggested.
Round 2
Reviewer 2 Report
The authors have provided reasonable responses. Thank you.